# Marker Assisted Introgression of Resistance Genes and Phenotypic Evaluation Enabled Identification of Durable and Broad-Spectrum Blast Resistance in Elite Rice Cultivar, CO 51

**DOI:** 10.3390/genes14030719

**Published:** 2023-03-15

**Authors:** Thiyagarajan Thulasinathan, Bharathi Ayyenar, Rohit Kambale, Sudha Manickam, Gopalakrishnan Chellappan, Priyanka Shanmugavel, Manikanda B. Narayanan, Manonmani Swaminathan, Raveendran Muthurajan

**Affiliations:** 1Department of Plant Biotechnology, Centre for Plant Molecular Biology and Biotechnology, Tamil Nadu Agricultural University, Coimbatore 641003, India; 2Department of Rice, Centre for Plant Breeding and Genetics, Tamil Nadu Agricultural University, Coimbatore 641003, India

**Keywords:** rice, biotic stress, blast, resistance genes, marker assisted selection, gene pyramiding

## Abstract

Across the globe, rice cultivation is seriously affected by blast disease, caused by *Magnaporthe oryzae*. This disease has caused heavy yield loss to farmers over the past few years. In this background, the most affordable and eco-friendly strategy is to introgress blast-resistant genes from donors into elite rice cultivars. However, it is not only challenging to evolve such resistance lines using conventional breeding approaches, but also a time-consuming process. Therefore, the marker-assisted introduction of resistance genes has been proposed as a rapid strategy to develop durable and broad-spectrum resistance in rice cultivars. The current study highlights the successful introgression of a blast resistance gene, i.e., *Pi9*, into CO 51, an elite rice cultivar which already has another resistance gene named *Pi54*. The presence of two blast resistance genes in the advanced backcross breeding materials (BC_2_F_2:3_) was confirmed in this study through a foreground selection method using functional markers such as NBS4 and Pi54MAS. The selected positive introgressed lines were further genotyped for background selection with 55 SSR markers that are specific to CO 51. Consequently, both *Pi9* as well as *Pi54* pyramided lines, with 82.7% to 88.1% of the recurrent parent genome recovery, were identified and the selected lines were evaluated under hotspot. The analysis outcomes found that both the lines possessed a high level of resistance against blast disease during the seedling stage itself. In addition to this, it was also noticed that the advanced breeding rice lines that carry *Pi9* + *Pi54* were effective in nature and exhibited a higher degree of resistance against blast disease compared to the lines that were introgressed with a single blast resistance gene. Thus, the current study demonstrates a rapid and a successful introgression and pyramiding of two blast resistance genes, with the help of markers, into a susceptible yet high-yielding elite rice cultivar within a short period of time. Those gene pyramided rice lines can be employed as donors to introgress the blast-resistant genes in other popular susceptible cultivars.

## 1. Introduction

Blast disease, one of the most devastating fungal diseases that affects the rice production system across the globe, is caused by the fungus *M. oryzae*. The disease accounts for about 70–80% of yield losses in rice. Blast-disease-infected plants exhibit stunted growth as it mainly infects the leaves, culm and panicles of the rice. The disease ultimately reduces the photosynthetic efficiency of the plants and its rice yield. The blast fungal pathogen produces lesions on the surface of leaves while it can infect the plant during any growth period of the crop (from seedling (nursery) to grain-filling stage). As a result, it reduces the rice production globally, causes heavy yield loss up to 70–80% and also affects the grain quality [1]. The incidence of blast disease is predominantly high in those areas with high humidity and low temperature. These conditions ultimately limit the rice yield and grain quality. Historically, it was first reported as a rice fever disease in China in 1637, and later, it was noticed in India, Korea and The Philippines [2].

In general, a chemical-based controlling method is not recommended as an effective approach for blast disease since it is highly expensive, hazardous to the environment and poses serious health problems to mankind. Conventional breeding techniques to develop breeds with resistance against blast disease are more time-consuming, tedious, require a greater number of generations and are more environment-dependent than the molecular breeding techniques. In recent times, advanced plant breeding methods have been proposed, such as the marker-assisted selection or marker-assisted backcross breeding method and gene pyramiding approach. These techniques achieve high throughput, are cost-effective and are also viable alternatives for conventional breeding methods since it enables the rapid introgression of blast resistance genes into susceptible rice varieties. Further, these techniques also enable the pyramiding of multiple resistance genes into a single rice variety, which can in turn contribute to the evolution of durable and broad-spectrum resistance against blast disease.

The majority of the high-yielding rice cultivars are susceptible to blast disease during wet seasons [3]. Though blast disease can be controlled using fungicides, it is not considered as an effective method since it incurs high costs and is hazardous in nature. Alternatively, the most economical and an ecofriendly approach is to explore the host plant resistance (R) gene that limits the incidence of blast disease [4]. Resistance to blast disease is generally governed either by major (which provide complete protection against a few variants of *M. oryzae*) or minor (which ensure partial protection against the pathogen) R genes [5].

Though conventional breeding methods have developed several blast-resistant rice cultivars, the linkage drag phenomenon has resulted in the inheritance of undesirable genes, which heavily confine the breeding process. Recent developments made in the identification and characterization of the R genes have revolutionized the breeding strategies which aim at the evolution of blast-resistant rice cultivars. Until this date, more than 500 QTLs associated with blast resistance have been reported, whereas 102 R genes have been mapped in rice for blast resistance. Among these 102 R genes, 38R genes, viz., *Pib*, *Pb1*, *Pita*, *Pid3–A4*, *Pikh*, *Pish*, *Pik*, *Pikp*, *Pi9*, *Pi2*, *Pizt*, *Pid2*, *Pi33*, *Pii*, *Pi36*, *Pi37*, *Pikm*, *Pit*, *Pi5*, *Pid3*, *Pia*, *PiCO39*, *Pi25*, *Pi1*, *pi21*, *Pi50* and *Pi65(t)* have been cloned and functionally validated at a molecular level [6]. The molecular breeding techniques supplement the routine blast-resistant rice breeding strategies by empowering a rapid and an accurate introgression of the target R genes [7] and successful introgression of the R gene into elite rice cultivars. Such hybrids have been well documented by employing the marker-assisted backcross breeding (MABB) approach [8,9,10].

Therefore, pyramiding multiple blast resistance genes into a single rice cultivar is an economical and an effective approach to widen or extend the spectrum of resistance in genotypes and to realize an enhanced and durable resistance to blast [11]. Further, it can also majorly reduce the cost of rice cultivation [12].

The majority of the high-yielding rice varieties are highly susceptible to blast disease, owing to the fact that the *M. oryzae* population has multiple variants. In addition to this, the frequent emergence of new virulent races also leads to the loss of resistance, even in resistant varieties within a span of 3–4 years. Hence, such resistant lines become susceptible to infection [13]. Therefore, it is necessary to combine broad-spectrum and durable resistance genes into a single elite rice cultivar. In the literature [14], researchers documented the large array of successful broad-spectrum resistance and effective blast disease resistance genes used so far in rice breeding programs. Among the existing genes, a minimum of 14 genes, viz., *Pi1*, *Pi2*, *Pi9*, *Pi20(t)*, *Pi33*, *Pi39* and *Pi40(t)*, *Pi 47*, *Pi48*, *Pi54rh*, *Pi56 PiZ*, *Piz-t* and *Pigm* have been reported for their broad-spectrum resistance [15,16]. Thus, the identification of the markers that are linked to the target trait and their application in the MABB technique help in the development of improved rice cultivars [17]. In particular, the gene pyramiding of the resistance genes or QTLs, through a marker-assisted selection process, has mitigated susceptibility and created an opportunity to develop resistance for the devastating blast disease in elite rice cultivar/genotypes [10,18]. Further, it also offered a practical solution to alleviate the issues pertaining to the breakdown of resistance [8].

*Pi9* is a major resistance gene that provides a broad-spectrum resistance against different variants of *M. oryzae* [19]. Among the blast resistance genes, *Pi54* is widely used by a number of researchers in breeding programs [18,20]. The variants of *Pi54* alleles, found among the wild rice species and other rice cultivars, exhibit numerous sequence variations. Such variants are hypothesized to be responsible for its broad-spectrum resistance feature [21,22,23]. A comparative study on the resistance mechanisms among the rice population cumulatively explained that the *Pi9* gene showed 52% resistance and *Pi54* gene exhibited 43% resistance against *M. oryzae* [24]. According to the literature, instead of singe gene introgression, the introgression of multiple genes provides a high level of durable resistance against blast disease. As per the All India Coordinated Rice Improvement Project (AICRIP) (DRR progress report, Vol.2, 2008–2013) report, both Swarna and Sambamahsuri NILs, containing the *Pi54* gene, recorded a high level of resistance in different rice cultivating regions of India. Therefore, along with the *Pi54* gene, the authors introgressed the *Pi9* gene into CO 51 to enhance its resistance spectrum and its durability against blast disease.

Lalat [25] and Tapaswini [26] reported a high level of resistance against blast disease in India through the introgression of *Pi2* and *Pi9* resistance genes into improved elite rice varieties. Similarly, the introgression of the *Pi9* resistance gene, along with other genes, into the Swarna rice variety displayed a Resistant (R) to Moderate Resistant (MR) reaction against blast disease [27]. Likewise, two-gene (namely, *Pi54* and *Pi1*) pyramided lines [28] and three-gene (namely, *Pi1*, *Pi2* and *Pi33*) pyramided lines [12] displayed high level of resistance reactions against blast disease. A combination of *Pi54* and *Pi1* genes in Swarna, Swarna-Sub1, Pusa basmati 1 and Sambamahsuri displayed heavy resistance to blast disease [29,30,31].

CO 51, an elite rice cultivar, is predominantly cultivated in the southern regions of India, especially in Tamil Nadu. However, the cultivar expresses moderate resistance to blast disease, as it has only *Pi54* [32]. In order to develop a superior blast-resistant version of this cultivar, CO 51 was introgressed in the study conducted earlier [33] with *Pi9*, using the donor 562-4 (a near isogenic line (NIL) of CO43) based on the marker-assisted backcross breeding (MABB) approach. Gene-based functional markers that are tightly linked with blast-resistant genes were already identified and utilized in MABB in rice. The gene pyramiding of the key resistance genes, with the help of marker-assisted selection, has resolved the problem of susceptibility for blast fungal disease [8]. In the current study, the cultivars were genetically enhanced to exhibit resistance against the blast disease by following two backcrosses, two generation selfing with stringent foreground selection and SSR-based background selection.

## 2. Materials and Methods

### 2.1. Plant Materials

CO 51 is a high-yielding fine-grain rice variety with a short duration (110–115 days) and is suitable for year-round cultivation in Southern India. The donor genotype, i.e., 562-4, is a NIL of CO 43 and it harbors a blast resistance gene, i.e., *Pi9* [33]. In this study, a breeding scheme was followed to develop advanced backcross breeding materials while the markers used for the selection of these materials, at different levels, are shown in Figure 1.

### 2.2. DNA Extraction

The authors extracted the genomic DNA from young fresh leaves of each advanced backcross progeny, using the CTAB method [34]. The quantity of the extracted DNA was measured using a Nanodrop ND-1000 spectrophotometer (Thermo Fisher Scientific, Wilmington, DE, USA).

### 2.3. Foreground and Background Selection

The primers flanking the *Pi9* on chromosome 6 (NBS 4F 5′-ACTTTGTTGTGCTTGATAAC-3′ and NBS 4R- 5′-ATGGTGAACGGTATCTGTAT-3′) [19] and *Pi54* on chromosome 11 (Pi54MAS F 5′-CAATCTCCAAAGTTTTCAGG-3′ and Pi54MAS R 5′-GCTTCAATCACTGCTAGACC-3′) [35] were utilized for the selection of target genes. A total of 55 polymorphic SSR markers (Appendix A) [36], spanning across the 12 rice chromosomes, was used for background analysis to survey the genome of the recurrent parent.

### 2.4. PCR Amplification

PCR was conducted in 15 μL reactions, containing 50 ng of the DNA template, 8.0 μL of sterile water, 1.5 μL of assay buffer (10×), 0.50 μL of 2.5 mMdNTPs, 0.20 μL (3 U/μL) of *Taq* DNA polymerase and 1.00 μL each of 10 μM forward and reverse primers that are specific to functional markers linked with the target genes, i.e., *Pi54* and *Pi9* [16]. PCR amplification was conducted with a profile of 35 cycles at 94 °C for 5 min, i.e., initial denaturation, 94 °C for 1 min denaturation, 55 °C for 1 min, i.e., annealing, 72 °C for 1 min extension, 72 °C for 10 min final extension and 4 °C to hold.

### 2.5. Genotyping and Data Analysis

The PCR products were resolved by 3% agarose gel electrophoresis using 1× TBE buffer and stained with Ethidium Bromide. Then, the gel was observed and documented using the standard gel documentation system (BIO-RAD, Hercules, CA, USA) and scored. The genotypic data, obtained from the mapping population, was analyzed using the GGT software (v2.0, Ralph Van Berlod Laboratory of Plant Breeding, Wageningen, The Netherlands). Subsequently, the graphical genotyping image was generated as described by Young and Tanksley (1989) [37].

### 2.6. Phenotypic Screening against Blast Disease

The plant materials were evaluated based on the Uniform Blast Nursery (UBN) method [38] at the Hybrid Rice Evaluation Centre (HREC), Gudalur, Ooty, Tamil Nadu (11°30′ N latitude; 76°30′ E longitude and 1317.00 m above MSL elevation) from February to March 2020. This location is a natural hot spot for screening the leaf blast disease resistance since the blast incidence is recorded throughout the year and the incidence becomes severe during winter and rainy seasons [39]. All the plant materials, including the parents (CO 51 and 562-4), R gene(s)-advanced backcross bred rice lines and the susceptible check (CO 39), were sown in a 100 cm row with 10 cm gap between the successive rows with three replications. The susceptible check CO 39 was sown in a single row after every six test entries. Furthermore, the entire UBN bed was surrounded by a single row of CO 39, as it enabled the easy spread of the blast fungal pathogen (Appendix A). Nitrogen fertilizers were applied to the culture beds since it increases the blast infection rate. The first dose of nitrogen (N) fertilizer was applied at the recommended rate 200 g/cent as a single pre-flood application (after 21 days of sowing). The second dose of the N fertilizer, i.e., 1.5 times of the recommended rate or 350 g to 400 g/cent was also applied as a single pre-flood application (after 35 days of sowing).

Scoring was conducted based on the IRRI Standard Evaluation System (SES) and three readings were recorded, starting from 30-day-old seedlings at 7–10 day intervals [40,41]. The test entries, with scores such as 0 and 1, were considered to be Highly Resistant (HR) and Resistant (R), respectively; those entries with a score of 3 were designated as Moderately Resistant (MR); entries with a score of 5 were considered as Moderately Susceptible (MS); entries with a score of 7 were classified as Susceptible (S) and finally, those entries with a score of 9 were regarded as Highly Susceptible (HS).

## 3. Results

The outcomes of the initial parental polymorphism analysis established the existence of polymorphism between the investigated parents, i.e., CO51 and 562-4, for the functional markers NBS4 [16] and Pi54MAS which were linked to the target, i.e., two blast resistance genes, viz., *Pi9* and *Pi54*, respectively (Figure 2).

### 3.1. Introgression of the Pi9 Gene into CO 51 and Foreground Selection

Initially, CO 51 and 562-4 were crossed against each other and F_1_ plants were subjected to foreground selection with the help of functional markers such as NBS4 and Pi54MAS. Subsequently, the F_1_ plants that harbor both *Pi9* and *Pi54* were backcrossed with CO 51. On the whole, 15 BC_1_F_1_ plants were raised and 10 positive plants were identified based on a foreground selection method. The positive BC_1_F_1_ plants were further backcrossed with CO 51 to generate the BC_2_F_1_ plants. A total of 228 BC_2_F_1_ plants was raised and screened for the presence of both blast-resistant genes with the help of the above-mentioned functional markers. Among the 228 progenies of the BC_2_F_1_ population, a foreground selection method was applied. Out of the total, 65 positive homozygous plants were obtained with both *Pi9* and *Pi54* gene combinations and 17 positive homozygous plants were arrived at, with the *Pi9* gene alone. Finally, 146 positive homozygous plants were found with only the *Pi54* gene.

Among the 65 positive homozygous BC_2_F_1_ plants with both the genes, line number # 5, which exhibited on-par performance with the recurrent parent (CO 51), was selected for further experiments and selfed to produce the BC_2_F_2_ plants. As a result, 14 BC_2_F_2_ lines were arrived at, containing both the blast resistance genes, *Pi9* and *Pi54*. On the other hand, six lines were found to contain a single blast resistance gene, i.e., *Pi54* only, whereas two lines were found to contain the other blast resistance gene, i.e., *Pi9* only, as shown in Table 1 (Supplemental Appendix A). Those lines were also subjected to background selection using 55 polymorphic SSR markers.

### 3.2. Background Selection

SSR-based background selection was performed to identify the *Pi9* and *Pi54* pyramided lines with the maximum recovery of the recurrent parent genome. In total, 55 SSR markers that exhibited polymorphism between the recurrent and the donor parents were used for this purpose. The recurrent parent recovery (RPG) was in the range of 82.7% to 88.1% in the advanced backcross breeding lines (Table 1). Two-gene pyramided lines and monogenic lines with *Pi9* or *Pi54* were selected and utilized for the evaluation of the phenotypic blast screening in the nursery field.

The graphical gene pyramided line in BC_2_F_2_ displayed that chromosomes 1, 2, 3, 6 and 12 of the lines contained heterozygous regions and different regions of genomic introgressions from 562-4. The CO 51 genome was completely regained in chromosomes 4, 5, 9 and 10 in all the BC_2_F_2_ lines. Chromosomes 6 and 11 displayed genomic introgressions from donor parent 562-4 while they also exhibited a maximum heterozygous region, due to the presence of *Pi9* and *Pi54* resistance genes in these chromosomes (Figure 3). This clearly indicates that the *Pi9* genes were introgressed into BC_2_F_2_ advanced backcross breeding lines from donor parent 562-4.

### 3.3. Evaluation of the Parents and the Advanced Backcross Population at Epiphytotic Locations

The evaluation outcomes of recurrent parent CO 51 (containing *Pi54* alone) and donor parent 562-4 (harboring *Pi9* alone) under natural hot spot conditions showed a moderate resistance against the blast pathogen in natural field conditions (Figure 4). Therefore, it can be speculated that the introgression of *Pi9* into the elite cultivar CO 51 increases the level of blast resistance.

The selected BC_2_F_2:3_ plants with two-gene combinations and single gene lines were also screened under natural hot spot conditions. The screening results against the blast pathogen revealed that both recurrent parent CO51 as well as donor parent 562-4 showed a MR reaction with a score of 3 and the susceptible check CO 39 showed a HS reaction with a score of 9. On the other hand, the gene-pyramided lines such as # 5-3, # 5-7, # 5-20 and # 5-21 with both the resistance genes, *Pi9* and *Pi54*, portrayed a high degree of disease resistance reaction with an R score of 1 against the blast disease (Table 1, Figure 5).

### 3.4. Performance of Rice Lines, Introgressed with Pi9, a Single Blast Resistance Gene

When screened under natural hot spot conditions for blast disease, the lines, such as # 5-11 and # 5-12, harboring single resistance gene *Pi9*, exhibited a MR reaction (Score = 3) against the blast disease (Table 1, Figure 5). It is evident from the outcomes of the blast screening experiment that the monogenic lines that harbor the *Pi9* gene alone exhibited a moderate level of resistance against blast disease.

Based on the results obtained from the blast screening experiment, it can be inferred that the gene pyramided lines that harbored both *Pi9* and *Pi54* resistance genes exhibited higher degrees of broad-spectrum resistance against blast disease. However, the monogenic lines that harbored only the *Pi9* resistance gene exhibited a moderate level of resistance against blast disease.

### 3.5. Performance of the Rice Lines, Introgressed with Pi54, a Single Blast Resistance Gene

The blast screening results confirmed that lines # 5-2 and # 5-10, harboring a single resistance gene *Pi54*, exhibited a Moderate Resistance reaction (MR) (with a score of 3) against blast disease (Table 1, Figure 5). Therefore, it is evident from the blast screening experiment results that the monogenic lines, harboring only the *Pi54* gene, exhibited a moderate level of resistance against the blast disease.

Based on the blast screening experimental results, it can be inferred that the gene pyramided lines, harboring both *Pi9* and *Pi54* resistance genes, exhibited higher degrees of resistance. It has the capability to contribute towards broad-spectrum and durable resistance against different strains of the blast fungus pathogen in different rice-growing regions of India. On the other hand, the monogenic lines that harbored only the *Pi54* resistance gene exhibited a moderate level of resistance against the blast disease.

## 4. Discussion

Among the prevalently reported rice diseases, blast fungal disease ranks first in terms of limiting the productivity of rice in India and other Asian countries. The disease has high incidence levels and records wide distribution during the growth periods of the rice plant under stable conditions. The introgression of multiple blast R genes that confer resistance to different pathogenic races has been identified as a strong tool in developing durable and broad resistance among elite rice cultivars [8]. Various studied have inquired about numerous blast-resistant genotypes and identified 100 blast R genes that provide resistance against *M. oryzae* infection [42]. The *Pi54* blast resistance gene provides a broad-spectrum resistance for various strains of *M. oryzae* [43].

The blast donor R genes that are mainly used in molecular breeding studies are mainly alleles or tightly linked R genes from three loci: *Piz*, *Pik* and *Pi-ta*. In the current study, the authors have pyramided the *Pi9* and *Pi54* blast R genes into an elite rice cultivar, CO 51. The *Pi9* gene can be located at the *Piz* locus whereas the *Pi54* gene is situated in the *Pik* locus. The rationale might be as follows: the first reason is the alleles or tightly linked R genes from these loci often provide relatively broad-spectrum resistance, especially against the leaf blast resistance during the seedling stage. Numerous R genes in other loci exhibit a narrow resistance spectrum or low-resistance effect which ultimately results in negative outcomes in terms of resistant improvement. The second reason is that some of the R genes’ linkage drag brings a negative effect on the agronomic traits. This phenomenon in turn restricts their usage in breeding practices [14].

Although various donors are available for blast resistance, the current study used 562-4 as a donor (with the *Pi9* gene), since it provides a better level of resistance against blast disease under field conditions in the target environment. The present study implemented two backcrosses to effectively introgress the *Pi9* resistance gene into CO 51 (which has the *Pi54* gene) followed by two generations of selfing, as displayed in Figure 1. A total of 22 BC_2_F_2_ generation lines of CO 51 and 14 homozygous plants for both the genes (*Pi9* + *Pi54*) were further subjected to foreground selection using the markers that are specific to the recurrent parent. Two lines which had a single gene *Pi9* and six lines which had a single gene *Pi54* were also selected to test the efficacy of both the genes in imparting blast resistance (Appendix A).

The main aim of the current study is to strengthen the plant defense mechanism or enhance the resistance against blast with the help of the marker-assisted method. At the same time, the authors ensured that the experiments are conducted without affecting the recurrent parent genome. Thus, it was important to characterize the gene-introgressed lines and select those lines which exhibited the highest recurrent parent genome recovery. The background selection was conducted using 55 SSR markers with genome-wide distribution and the RPG recovery rate was found to be in the range of 82.7% to 88.1% in BC_2_F_2_ generation (Table 1). The studies conducted earlier in rice reported similar kinds of RPG recovery in BC_1_F_1_ plants; for example, 71–79% by Basavaraj et al., in the year 2010 [44].

The graphical genotyping of the BC_2_F_2_-gene-pyramided lines confirmed that chromosomes 1, 2, 3, 6 and 12 of the selected lines had heterozygous regions along with different regions of genomic introgressions from 562-4. It was also observed that the CO 51 genome was completely regained in chromosomes 4, 5, 9 and 10 in all the BC_2_F_2_ lines. Chromosomes 6 and 11 displayed genomic introgressions from donor parent 562-4. Further, it also had the maximum number of heterozygous region, which might be due to the presence of *Pi9* and *Pi54* resistance genes in those chromosomes. Therefore, the background selection process suggests that *Pi9* was introgressed in the BC_2_F_2_ lines from 562-4 and it retained the rest of the genomic portions of CO 51, including *Pi54*.

In the current study, CO 51-pyramided lines that harbored both *Pi9* and *Pi54* genes displayed a score of 1 as their resistant reaction. However, the monogenic lines displayed a Moderately Resistant reaction with a score of 3 (Table 1). Therefore, the study results infer that the combinations of genes *Pi9* + *Pi54* in the pyramided lines (viz., # 5-3, # 5-7, # 5-20 and # 5-21) are highly effective in imparting the resistant reactions; the upregulation of the transcripts involved in Jasmonic Acid (JA) biosynthesis was identified in the NILs harboring *Pi9* and *Pi54* genes after *M. oryzae* infection [24]. In the literature, an alteration in the expression of JA was predicted to act as a strong mediator of resistance against blast disease [45]. On the other hand, the monogenic lines, either with *Pi9* (line # 5-11 and # 5-12) or *Pi54* (line # 5-2, # 5-10), displayed a moderate resistance reaction to blast disease. The introgression of the major R gene broad spectrum *Pi54* into Taipei 309, a blast-susceptible japonica rice genotype, increased the latter’s blast resistance by Callose deposition in the host cells [43]. The genetic enhancement of the elite rice cultivar CO 51, in terms of blast resistance, confirms the application of the *Pi9* + *Pi54* gene combination in the hotspot regions of southern India against blast disease.

On the other hand, it should also be noted that a few two-gene pyramided lines, such as #5-8 and #5-16 (with both *Pi9* and *Pi54*), have displayed a moderate level of resistance against blast disease instead of providing resistance. The probable reasons may be due to: (i) non-additive gene effects in #5-8 and #5-16 as reported in [46]; (ii) deployment of other loci required to ensure a R reaction (It has been shown that interaction effects between these target R genes (*Pi9* and *Pi54*) or different broad spectrum R genes may lead to different resistance level in these lines [47,48]); and (iii) recombination between the gene and markers used in this study or due to admixtures during generation advancement [26]. Hence, additional studies are required to sort out such a moderate level of resistance imparted by these genes.

Though the current study did not evaluate the BC_2_F_2_-pyramided lines of CO 51 in terms of yield and other quality traits, the authors’ previous reports on the marker-assisted gene-pyramiding of blast genes in rice exhibited an enhanced resistance towards blast disease, while retaining the same agronomic performance, yield and grain quality of the recurrent parent [47]. As per the literature, pyramiding multiple or more than one blast resistance genes in 07GY31 japonica rice genotypes showed no impact on the performance of the basic agronomic traits [48]. Although evidence has demonstrated that the interaction between *Pi9* and *AvrPi9* (as well as *Pi54* and *AvrPi54*) mediates a broad-spectrum resistance to rice blast disease [49], such evidence is yet to be demonstrated in advanced breeding materials. Hence, elaborate and extensive laboratory and field experiments should be conducted in the future to examine the resistance spectra of *Pi54* and *Pi9* genes against various strains of *M. oryzae*.

Resistance breakdown can be observed against blast disease in many rice-growing areas and the introgression of more than one gene into a single rice cultivar can mitigate the resistance breakdown rate. Further, it can also provide durable resistance against blast disease [8]. Genes *Pi9* and *Pi54* mapped on to Chromosomes 6 and 11, respectively, have been reported to confer a broad-spectrum resistance against blast disease [6]. Therefore, the *Pi9* + *Pi54* gene combinations used in this study can contribute to a broad spectrum and durable resistance against various strains of the blast fungal pathogen in CO 51, an elite rice cultivar.

## 5. Conclusions

A systemic approach to develop rice genotypes with durable and broad-spectrum resistance against blast disease depends on pyramiding multiple genes into susceptible rice genotypes. The current study outcomes suggest that the gene functional markers (namely, *Pi9* and *Pi54*) are promising and can be used as potential markers to introgress blast resistance. With this, the accelerated generation of the improved rice line can be achieved in a short period of time through a MABB approach. The current study took such an effort which helped in the development of BC_2_F_2:3_ lines in CO 51, an elite rice variety. The lines exhibited a high degree of resistance to blast disease. Therefore, the current study demonstrated that the introgression of two blast resistance genes can contribute to a broad-spectrum or durable resistance against different strains of the blast fungus pathogen. Further, the improved version of CO 51 can serve as a useful genetic resource in future breeding programs that aim at introgressing blast resistance in other popular rice varieties and can serve as a useful base rice material for host–pathogen interaction studies.

## Figures and Tables

**Figure 1 genes-14-00719-f001:**
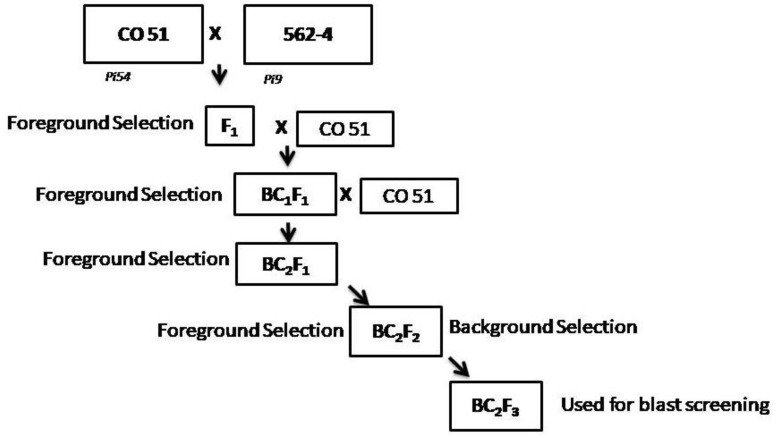
Breeding scheme used in this study for the development of blast-resistant version of CO 51.

**Figure 2 genes-14-00719-f002:**
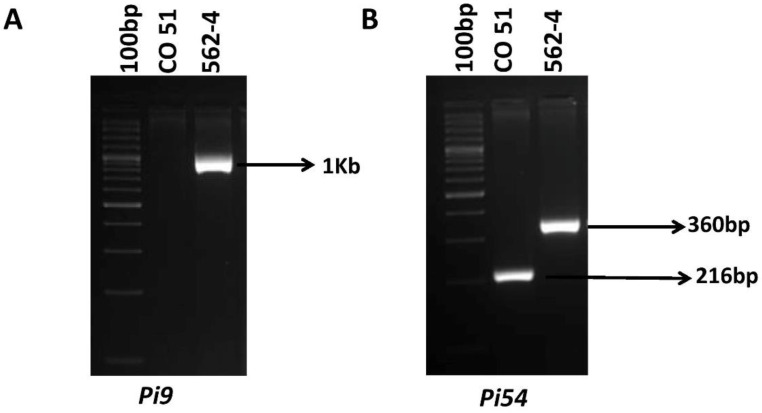
Parental polymorphic survey with the target genes. (**A**) PCR amplification of the NBS4 marker linked with *Pi9* gene. Allele of *Pi9* (1 kb) was found only in the 562-4 (donor line). (**B**) PCR amplification of the Pi54MAS marker linked with *Pi54* gene. The resistance (216 bp) and the susceptible (359 bp) alleles, produced by *Pi54* [16], are shown in the image with respect to CO 51 (recurrent parent) and 562-4 (donor parent).

**Figure 3 genes-14-00719-f003:**
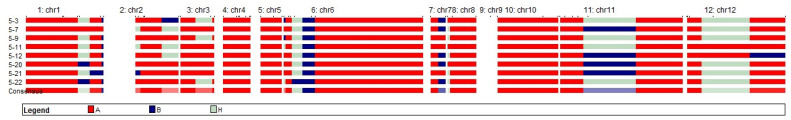
Graphical genotyping of the advanced backcross lines. The red color indicates the homozygous regions for CO 51 (recurrent parent). The blue color indicates the homozygous regions for 562-4 (donor parent) and the light green color indicates the heterozygous regions.

**Figure 4 genes-14-00719-f004:**
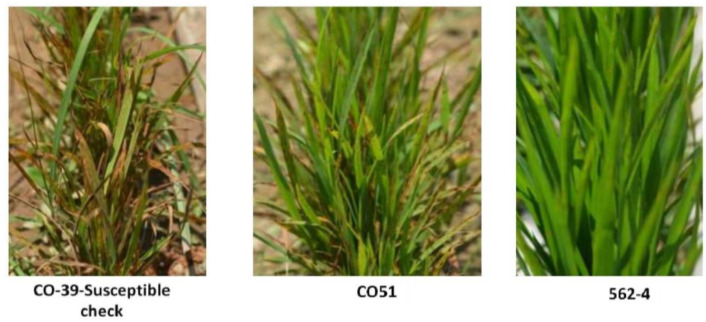
Response of the rice genotypes such as CO 39 (susceptible check), CO 51 (recurrent parent) and 562-4 (donor parent) to the blast disease.

**Figure 5 genes-14-00719-f005:**
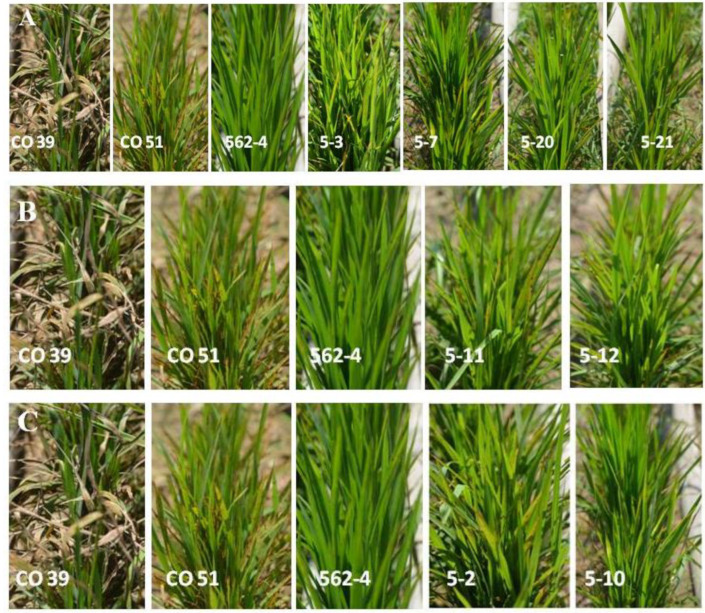
(**A**) Phenotypic responses of the susceptible check (CO 39), recurrent parent (CO 51), donor parent (562-4) and double genes (*Pi9* and *Pi54*)-introgressed lines under uniform blast nursery (**B**) Phenotypic responses of the susceptible check (CO 39), recurrent parent (CO 51), donor parent (562-4) and single gene (*Pi9*)-introgressed lines under uniform blast nursery (**C**) Phenotypic responses of the susceptible check (CO 39), recurrent parent (CO 51), donor parent (562-4) and single gene (*Pi54*)-introgressed lines under uniform blast nursery.

**Table 1 genes-14-00719-t001:** Details of the BC_2_F_2_ lines that harbor the blast resistance genes, *Pi9* and *Pi54*.

S. No	Pyramid Lines	*Pi9*	*Pi54*	RPG * %	Disease Score	Reaction ^#^
1	CO 39 (Susceptible check)	−	−	-	9	HS
2	CO 51 (Recurrent parent)	−	*+*	-	3	MR
3	562-4 (Donor parent)	*+*	−	-	3	MR
4	5-1	+	+	-	1	R
5	5-2	−	+	-	3	MR
6	5-3	+	+	84.5%	1	R
7	5-4	−	+	-	3	MR
8	5-5	+	+	-	1	R
9	5-6	+	+	-	1	R
10	5-7	+	+	87.2%	1	R
11	5-8	+	+	-	3	MR
12	5-9	+	+	85.4%	1	R
13	5-10	−	+	-	3	MR
14	5-11	+	−	88.1%	3	MR
15	5-12	+	−	85.4%	3	MR
16	5-13	−	+	-	3	MR
17	5-14	−	+	-	3	MR
18	5-15	−	+	-	3	MR
19	5-16	+	+	-	3	MR
20	5-17	+	+	-	1	R
21	5-18	+	+	-	1	R
22	5-19	+	+	-	1	R
23	5-20	+	+	87.2%	1	R
24	5-21	+	+	83.6%	1	R
25	5-22	+	+	82.7%	1	R

+ indicates presence of gene, − indicates absence of gene, * RPG-Recurrent Parent Genome. ^#^: Susceptible; R: Resistant; MR: Moderately Resistant; HS: Highly Susceptible.

## Data Availability

Not applicable.

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
