# Peer review of "Marker Assisted Introgression of Resistance Genes and Phenotypic Evaluation Enabled Identification of Durable and Broad-Spectrum Blast Resistance in Elite Rice Cultivar, CO 51"

_genes, 2023, doi:10.3390/genes14030719_

Round 1
Reviewer 1 Report
General comments
The major concern is that having the two genes Pi9 and Pi54 does not equal to durable and broad-spectrum blast resistance, which is demonstrated by the lines 5-8 and 5-16. Therefore, an extensive rewriting must be done to tone-down the paper's results and conclusions (and the title of the article as well). The paper must refer specifically to the introgression of the Pi9 and Pi54 but refraining to equal the harboring of those two genes to having "durable and broad-spectrum" blast resistance.
The English language and style across the manuscript must be extensively revised. Some of the issues that stand out are:
1. The use or omission of the articles "the" or "a/an" seems arbitrary or different from which is the most common use in English. Just to point out a few cases:
- line 17 "using conventional breeding approach" should be "using conventional breeding approaches" (if there were different possible conventional approaches) or "using a conventional breeding approach" (if there were just one possible conventional approach, the "a" indefinite article is required).
- line 44 "Majority of" should be "The majority of", because "majority" is a noun;
- line 47 "plant resistance (R) genes is considered as most economical and ecofriendly 47 approach" should be "plant resistance (R) genes is considered as the most economical and ecofriendly 47 approach", because a superlative adjective is being used.
- line 39 "especially in the areas where high humidity and low temperature are prevailing" should be "especially in areas where...", because there is not a defined set of humid areas that can be called "the areas", but any rice area that meets these environmental conditions can suffer from the disease.
These same issues and other language errors (e.g. "Totally" instead of "In total" in line 141) are recurrent all over the manuscript and should be revised.
2. Typos and punctuation errors are prevalent and should be revised. Please refer to https://www.mdpi.com/authors/layout#_bookmark33 and attach to those rules.
Just to cite a few examples:
Lines 89 and 91. Closing parenthesis are missing;
Line 114, two consecutive opening parentheses;
Line 116, the space between number and unit is missing ("100cm" should be "100 cm"). ;
Introduction
The current and sanctioned name for the blast pathogen is Pyricularia oryzae, and so must be referred in the article. Please check:
Resolving the polyphyletic nature of Pyricularia (Pyriculariaceae), S. Klaubauf, D. Tharreau, E. Fournier, J.Z. Groenewald, P.W. Crous, R.P. de Vries, and M.-H. Lebrun, STUDIES IN MYCOLOGY 79: 85–120.
https://www.studiesinmycology.org/sim/Sim79/Resolving-the-polyphyletic-nature-of-Pyricularia-Pyriculariaceae-_2014_Studies-in-Mycology.pdf
Lines 39-40. It seems contradictory to state that blast disease has been reported "irrespective of the rice environments", and then immediately say that "especially in the areas where high humidity and low temperature are prevailing". Rice blast occurrence is not irrespective of rice environment.
Materials and Methods
My major concern regarding the paper's methodology is the lack of any formal statistical test to evaluate the association between the alleles of the functional markers and the blast resistance scores and/or the disease reaction. I encourage the authors to perform a proper statistical analysis including all the lines that were genotyped and tested for blast resistance during the backcross and introgression process (including the corresponding checks and parents).
Figure 1 and elsewhere: "Fore ground" and "Back ground" should be replaced by "Foreground" and "Background", respectively (without the separating space).
Line 78. The reference for the donor parent is wrong, it does not mention the genotype 562-4. Also, it is not in the format required by the journal. It must be provided a proper reference for the genotype 562-4 documenting which blast resistance gene(s) it has.
Line 88. Cite a reference for the Pi9 NBS4 marker.
Line 92. Provide a list of the 55 SSR markers (it may be included in the Supplementary file), and the corresponding citation(s).
Line 109. Provide a citation for the UBN method.
Line 119. Detail the dose and timing for the N fertilization.
Results
There major results are clearly presented. However, there are several concerns that must be addressed by the authors.
First of all is the conclusion that the observed blast resistance is "durable and broad-spectrum". There is no single study in the article that proofs this statement. The authors seem to equate the R reaction and a score of 1 in a to durable blast resistance, which is not necessary the case.
In lines 149-150, its stated that the BC2F1 Line #5, which was found to be homozygous for the two genes, "have shown on-par performance with the recurrent parent (CO 51)". Do you mean they had the same level of blast resistance? Or are you talking about agronomic performance? Authors must specify which performance are they referring here, and describe in the M&M section how the performance of the BC2F1 lines was evaluated and compared to that of the parents. If that "on-par performance" means that a homozygous line carrying both genes had the same blast resistance as the donor with only one of the genes, most of the reasoning in the article is contradicted, thus this finding should be properly addressed and discussed.
Other findings that are not mentioned by the authors must be pointed out in the Results section and then more extensively addressed in the Discussion section.
Discussion
Lines 229 to 280 belong to an Introduction section better than to a Discussion section.
Lines 322-325. Authors must deepen the discussion of this finding, analyzing: 1) possible causes and explanations for this particular result, each one with its corresponding citation, and 2) what this implies for the major results and conclusions of the paper. In particular, those two lines suggest that there are more blast resistance genes required for a 1 score and a R reaction than just Pi9 and Pi54. Thus, there is no total equivalence between being homozygous for Pi9 and Pi54 and having "durable and broad-spectrum" blast resistance. More importantly, those other genes could be lost during the recovery of the recurrent parent, leading to lines with intermediate blast resistance such as that exhibited by lines 5-8 and 5-16. Authors must rewrite their findings and conclusions under this light.
Reviewer 2 Report
1. Pyramiding multiple resistance genes to widen or extend blast resistance spectrum is a good tropic in molecular breeding.
2. The rice variety, Co 51, with Pi54 resistance gene and expressed moderate blast resistance. Many resistance genes could be selected to introgressed into Co51 and to further improve its blast resistance. In this study, why is Pi9 used instead of other genes? So, in the section “introduction”, the authors should describe how difference between Pi9 and Pi54 in disease resistance mechanism by citing more information (references), and clarify how these two genes are complementary in functionally, or blast resistance spectrum, and it is necessary to combine them.
3. Section “discuss” should be concise, and mainly describe the novelties, the problems, and views on future work. Some content, such as the line 246-273, 313-320 could be sketched in section “introduction”.
4. The MAS schedule involve in genetic background substitute to Co51, SSR maker evaluation showed that Pi9 and Pi54 pyramided lines of BC2F2 with 82.7% to 88.1% of recurrent parent genome. However, there is approximately 15% genome of “562-4”, which contains thousands of genes affecting the growth, development of rice, though many of those are same. Therefore, in addition to SSR marker evidence, data of some agronomic traits are needed to support that, when backcross to BC2F2, MAS approach can pyramid of two blast resistant genes to enhance the resistance to rice blast, while other important agronomic traits are similar or no worse to the recurrent parents.
5. The authors supple some evidences to elucidate the improving seedling blast resistance by pyramid Pi9 and Pi54, however, it is better to show the evidence of blast resistance at other growth stage, such as panicle blast resistance.
6. In table 1, the pyramided line 5-22 showed that it with both two resistance genes. However, in supplementary file, the Pi54 band was absent in 5-22 line in the gel.
7. In lines 304-305, the sentence “Therefore, it can be concluded that Pi9 and Pi54 genes were introgressed in BC2F2 lines from 562-4 and retained remaining genomic portions of CO 51” may cause confuse, as the Pi 54 was already in the Co51, instead of introgressed from 562-4.
8. In line 415, “Ning, X.; Yunyu, W.; Aihong, L.” should be “Xiao, N.ï¼›Wu, Y.; Li, A.”
Round 2
Reviewer 1 Report
Most of the minor and formal suggestions were satisfactorily addressed. The author's effort in improving the language and formal issues in the paper is acknowledged. The new version of the manuscript is much more readable and meets the appropriate formal aspects for publication.
However, the major concern pointed out in the first revision was not properly addressed. Your finding of the two lines (5-8 and 5-16) harboring the genes Pi9 and Pi54 and yet display a MR blast reaction demonstrates that these two genes (or, more properly, the two functional markers) are not enough to ensure, in the genetic background of CO 51, a R reaction to the blast isolates that are present in your UBN. The explanation you provided for this finding in the revised manuscript (i.e. downregulation of JA and low callose deposition) is not satisfactory.
Bringing up JA regulation and/or deposition of callose does not add up to the explanation of the results you found for lines 5-8 and 5-16. It is just the physiological pathway in which Pi9 and Pi54 are reported to lead to a blast resistance phenotype; upregulation leads to R and downregulation leads to S or MR. The real issue you must address is why some NILs harboring the two genes would upregulate and have R reaction, and why some others with the same alleles would downregulate and have MR reaction. You state (line 455 of the revised manuscript) that the NILs that have the two genes and are R and they upregulate JA. Then you say that there are NILs that also have the same two genes, but yet they are MR, so they downregulate JA. That is circular reasoning and it contradicts logical thinking. What is causing that these same two genes/markers do work in some NILs but do not work in some others? This is the real issue that you must address, and it was not addressed in the revised manuscript.
To my knowledge, there are only two possible explanations for this, and both must be addressed in the discussion (of course if the authors find any other logically and biologically valid explanation, should expose it too):
1) Pi9 and Pi54 are not sufficient for blast resistance. 1a) Either there must be other loci required to ensure R reaction that are present in some NILs but absent in 5-8 and 5-16, or 1b) there are other loci that interfere in some way with the effects of Pi9 and Pi54.
2) The functional markers used for selection have (even with very low frequency) some probability of recombination with the genes they are linked to. Then, lines 5-8 and 5-16 would only appear to harbor Pi9 and Pi54, but they do not, because there were some recombination events between the genes and their linked markers. The observed frequency of this supposed recombination in the total population of NILs should be calculated and compared with the expected frequency based in the physical distance between the gene and the marker.
All this possible explanations must be discussed and documented with their correspondent citations.
Without properly addressing the results you found for lines 5-8 and 5-16, the paper would be ignoring the scientific evidence produced by your own work, and thus would lack the scientific rigour required for publication.
Reviewer 2 Report
1. Line 253, “such as” could be deleted.
2. In figure 3, The red color indicates the homozygous regions for CO 51 (recurrent parent). The blue color indicates the homozygous regions for 562-4 (donor parent), and the light green color indicates the heterozygous regions. What do other colors, pink and purple represent?
3. Lines 286-287, the words “it can be concluded that Pi9 and Pi54 genes” should be deleted?
